# Measurable Residual Disease Detection in Acute Myeloid Leukemia: Current Challenges and Future Directions

**DOI:** 10.3390/biomedicines12030599

**Published:** 2024-03-07

**Authors:** Jennifer Moritz, Antonia Schwab, Andreas Reinisch, Armin Zebisch, Heinz Sill, Albert Wölfler

**Affiliations:** 1Division of Hematology, Medical University of Graz, 8010 Graz, Austria; 2Department of Blood Group Serology and Transfusion Medicine, Medical University of Graz, 8010 Graz, Austria; 3Division of Pharmacology, Otto Loewi Research Center for Vascular Biology, Immunology, and Inflammation, Medical University of Graz, 8010 Graz, Austria

**Keywords:** acute myeloid leukemia, measurable residual disease, risk-stratification, MRD detection methods, postremission decision

## Abstract

Acute myeloid leukemia (AML) is an aggressive malignant disease with a high relapse rate due to the persistence of chemoresistant cells. To some extent, these residual cells can be traced by sensitive flow cytometry and molecular methods resulting in the establishment of measurable residual disease (MRD). The detection of MRD after therapy represents a significant prognostic factor for predicting patients’ individual risk of relapse. However, due to the heterogeneity of the disease, a single sensitive method for MRD detection applicable to all AML patients is lacking. This review will highlight the advantages and limitations of the currently available detection methods—PCR, multiparameter flow cytometry, and next generation sequencing—and will discuss emerging clinical implications of MRD test results in tailoring treatment of AML patients.

## 1. Introduction

Acute myeloid leukemia (AML) is an aggressive hematologic malignancy characterized by clonal proliferation of hematopoietic stem and progenitor cells (HSPCs) due to genetic aberrations, leading to the uncontrolled growth of myeloid blasts that outcompete and displace normal hematopoiesis [1]. Prognosis and outcome varies widely depending on the underlying molecular and cytogenetic aberrations as well as factors such as the patients’ age and comorbidities [2,3,4,5]. Although treatment modalities have significantly improved over the last decade and most patients will achieve complete hematologic remission (CR) after intensive chemotherapy, the mean 5-year overall survival (OS) is lower than 50% [6]. The outcome in elderly AML patients (>65 years) is even worse, with a median 2-year survival of only 20% after diagnosis [7]. This low long-term survival rate in AML patients is due to high rates of relapse, which are caused by trace amounts of chemoresistant leukemic cells in the bone marrow (BM) that persist despite complete hematologic remission. These persistent cells are referred to as minimal or measurable residual disease (MRD) and act as potential precursors to relapse, which has raised high clinical interest in MRD testing in AML [8].

As in other hematological malignancies, such as chronic myeloid (CML) and acute lymphoblastic leukemia (ALL), MRD evaluation in AML serves various purposes: to offer a quantitative approach for determining the depth of remission, to enhance the assessment of relapse risk after achieving remission, and to detect signs of an impending relapse earlier allowing for immediate intervention [5,9]. Furthermore, MRD negativity represents a potential surrogate endpoint for clinical trials in predicting survival [10]. The prognostic relevance of MRD in AML has been documented by many studies; in a recent meta-analysis published by Short et al. comprising more than 80 studies, the overall 5-year survival of patients attaining MRD negativity after therapy was twice as high as in patients without (68 vs. 34%). The significance of MRD remained consistent regardless of age groups and AML subtypes [7]. Therefore, testing for MRD has emerged as a critical concept in the treatment of AML [11] (see Figure 1). Accordingly, current European LeukemiaNet (ELN) guidelines for AML diagnostics and therapy recommend MRD detection for all AML patients in complete remission after intensive chemotherapy to gain prognostic information on each patient’s risk of relapse [5,12]. However, while prognostic factors such as the age of AML patients and distinct cytogenetic and molecular aberrations at diagnosis are well established for therapeutic decisions, changes in MRD-guided treatment still lack robust results from randomized phase III trials to fully support standard clinical practice [5]. Furthermore, MRD evaluation in AML still encounters various challenges limiting its use in routine clinical testing. In contrast to other hematological malignancies such as ALL and CML, detection of MRD in AML is based on different methods not applicable to all patients, making comparison between different methods and clinical studies difficult [5,6,13]. Also, time points of testing as well as specimen source (peripheral blood (PB) vs. bone marrow (BM)) vary between studies and contribute to uncertainty regarding the widespread use of MRD testing in clinical practice and its potential adoption as an endpoint in clinical trials [10].

In this review, we will depict current methods of MRD detection in AML, focusing on their advantages and disadvantages. In addition, we will present novel concepts to overcome some of these limitations. We will discuss the current literature regarding clinical implications of MRD status and its impact on therapeutic decisions in intensive as well as non-intensive AML treatments. Finally, we will summarize the implications and shortcomings of MRD detection in standard clinical practice.

## 2. Current Methods of MRD Detection

MRD assays have the capability to identify persistent single leukemic cells among 10^3^ to 10^6^ normal hematopoietic cells, resulting in a sensitivity ranging from 1 × 10^−3^ to 1 × 10^−^⁶, providing additional insights into the depth of remission at various time points. The definition of MRD negativity thus depends on the applied MRD technology and target, but for all methods at least requires MRD levels below a detection threshold of 1 × 10^−3^ cells [14]. Currently, MRD detection in AML is either performed by multiparameter flow cytometry (MFC) analysis or quantitative polymerase chain reaction (qPCR)-based detection of leukemia-specific genetic aberrations [5,15]. Furthermore, detection of distinct leukemia-specific molecular aberrations by next generation sequencing (NGS) is becoming increasingly important for measuring MRD in AML [16,17]. A summary of the advantages and shortcomings of each MRD detection method is given in Table 1.

### 2.1. PCR-Based Methods

MRD detection by qPCR shows high sensitivity ranging from 10^−4^ to 10^−6^, and is therefore considered the gold standard. However, since only about 40–50% of patients show specific mutations and/or translocations that can be detected by qPCR, the method is not applicable to all AML patients [8,18]. A quantitative PCR test for MRD assessment is advised for patients with AML that display a consistent and distinct leukemia-specific genomic abnormality. This includes individuals with an *NPM1* mutation, core-binding factor leukemias displaying either a *RUNX1::RUNX1T1* or a *CBFB::MYH11* fusion gene, acute promyelocytic leukemia with a *PML::RARA* fusion gene, AML with *KMT2A::MLLT3* or *DEK::NUP214* gene fusions, and in very rare cases, *BCR::ABL* positive AMLs [5]. For those patients, qPCR offers excellent sensitivity, the advantage of being available in most laboratories, high sample throughput and fast turnaround times [19]. Persistence of the above-named mutations or fusion genes after chemotherapy detected by qPCR is a strong indicator of relapse, thus, ELN guidelines recommend specific time points for MRD assessment in such AML patients: at diagnosis of AML, after two cycles of chemotherapy, at the end of treatment (EOT), as well as every three months (BM) or four to six weeks (PB) during two years of the follow-up period. If patients undergo allogeneic stem cell transplantation (alloSCT), testing for MRD is also advised before the start of conditioning treatment, as MRD status is known to be a strong predictor of the post-transplant outcome [5,20,21,22]. ELN guidelines recommend PCR-based MRD analysis to be run in triplicate. The analysis is considered positive, if amplification can be confirmed in two of three replicates with Ct values lower than 40 [5].

Other than being applicable only to approximately half of all AML patients, qPCR also has the disadvantage of only allowing a relative quantification of targets in the analyzed sample. The Ct value offers insights into the initial quantity of the detected mutation or fusion transcript, but only in terms of the relative copy number. To overcome this limitation, the target is usually compared to a standardized transcript such as *ABL1* [6]. However, the use of different transcripts makes comparisons between different assays difficult. Furthermore, the same qPCR tests run in different laboratories, e.g., for testing mutated *NPM1*, may lead to discrepant results [23]. Although the sensitivity of MRD detection using PCR was high, Scott et al. were able to demonstrate an issue with *NPM1* analysis in an interlaboratory study. As part of the study, various AML samples being MRD high, low, or negative were sent to 29 different laboratories in 12 countries for MRD testing. MRD detection of *RUNX1::RUNX1T1*, *CBFB::MYH11*, *PML::RARA,* and *NPM1* was assessed by qPCR. The study revealed various testing errors, leading to a false-positive MRD result when analyzing an *NPM1* mutation negative AML sample in many of the participating laboratories. In comparison to the fusion transcripts analyzed by PCR, the *NPM1* type A mutation is caused by duplication, which makes it prone to a higher likelihood of false-positive results. During the process of reverse transcription before analysis, errors in *NPM1* exon 12 can be introduced artificially. As *NPM1* is regarded as a leukemia-specific mutation, falsely positive MRD results could lead to unnecessary treatment [23]. Therefore, current efforts undertaken by groups, such as the ELN–DAVID MRD Working Group, focus on standardization and quality control of qPCR-based MRD assays to generate comparable MRD results [24].

### 2.2. Flow Cytometry

For approximately half of patients lacking a specific mutation or fusion gene detectable with qPCR, the current recommendation to monitor MRD is to employ multiparameter flow cytometry (MFC) [5]. MFC allows the detection of aberrant antigen-expression in leukemic cells [25] by two different approaches that have been established for MRD assays. One approach centers on identifying leukemia-associated immunophenotypes (LAIP) with aberrant marker expression as compared to physiological HSPCs at the time of diagnosis, which are then monitored throughout and after treatment [26]. The second approach focuses on identifying antigen expression patterns of cell populations that are “different from normal” (DfN) [27]. In contrast to LAIPs, the latter method does not rely on a MFC analysis of the initial diagnostic leukemia sample. Due to changes in immunophenotypic patterns over the course of treatment, the LAIP approach could result in false-negative test results, while the DfN method has a higher risk for being positive, without truly detecting residual leukemic cells [28]. As a result, the ELN MRD Working Party has suggested the adoption of a combined approach referred to as the “LAIP-based DfN approach”. This approach entails the thorough integration of specific LAIP tracking within a comprehensive immunophenotypic profiling of BM cells [5].

Cytometers have the capacity to analyze a minimum of eight markers simultaneously and can process several million cells within minutes. According to the ELN recommendations, MFC-MRD panels are advised to comprise a fundamental set of surface markers, specifically early progenitor markers (CD34 and CD117), myeloid markers (CD11b, CD13, CD15, and CD33), along with various differentiation markers (CD2, CD7, CD19, CD56, and HLA-DR) to track AML cells with their aberrant marker expression. It is recommended to use CD45 gating with forward/side scatter plots, and for each sample at least 500,000 events should be recorded. To identify the MRD population, it is recommended to compare it with the diagnostic pattern at the time of diagnosis, looking for residual cells from the same population [29,30]. Numerous studies have demonstrated that the recognition of leukemic cells through MFC after induction or consolidation chemotherapy serves as a strong predictor of an elevated risk of relapse and impacts overall survival [28,31,32,33,34,35]. Loke et al. could furthermore show that post-transplant MRD positivity assessed by MFC is associated with significantly reduced overall survival [36]. One of the largest studies analyzing MRD results obtained by MFC, with over 1000 AML patients in CR after induction chemotherapy, showed that detection of MRD in CR was associated with statistically significant inferior survival. A cutoff of 0.1% was used to distinguish MRD-positive from -negative patients [28].

The advantages of this method include its wide applicability to almost all patients, limited costs, and the rapid availability of the results. However, the sensitivity rates compared to PCR are lower, at 10^−3^ and no standardized MFC method for MRD detection currently exists [37,38]. Differences in laboratory equipment, sample processing procedures, and cytometer configuration result in poor comparability between laboratories [39]. Also, the combinations of markers for AML MRD detection varies between different institutions, resulting in different operator-dependent gating strategies. Furthermore, all examiners require a high level of expertise in the interpretation of MFC-based MRD analysis [28,40]. Further limitations of the method include immunophenotypic shifts at relapse due to the possible evolution of subclones, which increases the difficulty of detecting MRD [37].

Ongoing efforts to enhance this methodology are centered on creating and validating antibody panels, which facilitate a standardized MFC-based MRD assessment, possibly also by automated methods for the analysis and interpretation of MFC-MRD data [6,41]. To make precise clinical decisions, it is crucial to employ a thoroughly validated and dependable assay that aligns with current regulatory standards. Tettero et al. developed a semi-quantitative MFC-MRD assay and published the analytical validation of their approach [42]. For validation, their method was compared to an alternative flow cytometry assay routinely used for the detection of hematological malignancies by comparing the percentage of LAIPs and blasts. The concordance was 0.99 for the percentage of blasts and 0.93 for the percentage of LAIPs between the two MFC approaches. Furthermore, they assessed the specificity and sensitivity and compared the results with their method between different operators (inter-operator precision), interpretation between different technicians (inter-gating precision), different FACS instruments (inter-instrument precision), and different laboratories. Also, they assessed the stability of the specimens and antibody reagents. Their results showed a high accuracy of their MFC-MRD assay to correctly quantify a LAIP at diagnosis and MRD at follow-up. However, the main limitation was the lack of a suitable reference assay. Also, variability may be introduced by differences in the gating strategy employed by individual operators and small discrepancies between samples that may arise from pipetting errors [42]. 

A future hope and possible advantage of this method is that MRD measurements via MFC from peripheral blood could yield results with similar sensitivity as analysis performed from BM. This would provide the advantage of less invasive procedures for AML patients during follow-up after treatment. Prospective, comparative assessments of this approach aiming to enhance the precision of PB vs. BM MFC-MRD testing are warranted [43,44]. The assessment of residual leukemic stem cells (LSC) using MFC-MRD is currently under investigation in clinical research [45,46]. Leukemic stem cells can be identified immunophenotypically as CD34^+^/CD38^low^ cells expressing aberrant markers such as CD45RA (PTPRC), CLL-1 (CLEC12A), or CD123 (IL3RA). LSC-MRD is expected to show greater sensitivity and reduced false-negative results as compared to standard MFC-MRD, showing another promising future perspective for MRD detection [5].

### 2.3. Next Generation Sequencing

Next generation sequencing using gene panels of interest offers an appealing approach for detecting measurable residual disease in AML [47]. The main advantage of this strategy is the applicability to a wider range of patients harboring different molecular aberrations in comparison to single-gene molecular testing [48]. The capacity to analyze an extensive number of genes with significant depth from both DNA and RNA has sparked significant interest in the utilization of NGS-based technology for MRD testing in AML [6]. Continued MRD positivity after treatment, as determined by the same NGS panel as employed during diagnosis, serves as a more sensitive biomarker for detecting persisting leukemic clones when compared with conventional non-molecular techniques. Furthermore, it has been proven to show prognostic value in predicting future relapse and mortality [49,50]. After the first consolidation therapy, MRD assessment by NGS was shown to predict relapse risk most accurately [51].

One of the specific targets for NGS-MRD, which has recently gained much interest, is the detection of *FLT3*-internal tandem duplications (ITDs) in AML patients [52]. Positive MRD assessment of NGS-based *FLT3*-ITD MRD in AML patients was identified as an independent risk factor for increased risk of relapse and inferior overall survival in various studies [53,54]. Tiong et al. showed that patient with detectable *FLT3*-ITD MRD by NGS (≥0.001%) had a significantly inferior 4-year overall survival rate and a higher cumulative incidence of relapse after alloSCT [55]. Other studies supported those results, also showing that positive MRD with NGS detected *FLT3*-ITD before alloSCT being highly predictive of relapse and inferior overall survival [56,57].

However, sequencing devices and especially short read sequencers pose the risk of base calling errors by calling incorrect nucleotides, which limits detection of mutations with low variant allele frequency (VAF), making NGS alone a potentially error-prone method for detecting MRD. Sequencing errors are commonly observed in roughly 0.1% to 1% of the sequenced bases. The error rates vary significantly depending on the type of mutation detected, with false positives occurring much more frequently in the detection of single nucleotide variants than in insertions or deletions [48,58]. To improve the sensitivity of NGS assays and allow the detection of mutations with a VAF < 1%, error-corrected sequencing has been introduced. Various complex methods for error-corrected sequencing exist, including the determination of a baseline error rate for every position within the specific region of interest, by comparing nucleotides in a single position between healthy individuals. The frequency of a falsely detected base in healthy individuals can provide insight into whether the detected alteration in the patient is truly pathogenic or just presenting background error [58,59]. Another way to improve the sensitivity of MRD-NGS analysis is to compare the remaining post-treatment-mutations detected by NGS with mutations initially present at the time of diagnosis. For example, if a *TP53* mutation was found at the initial diagnosis of AML, the detection of a *TP53* mutation with a very low VAF < 1% after therapy is most likely still representing a correctly positive result [60]. Another strategy in error-corrected sequencing entails the inclusion of random oligonucleotides or unique molecular identifiers (UMIs) during the library preparation step before DNA amplification. This process enables the tagging of individual DNA molecules with a distinct molecular fingerprint [48]. Dillon et al. were able to show that when using duplex sequencing (DS), an ultrasensitive NGS method that generates double-stranded sequences, false positive errors can be reduced, and the prediction of relapse and survival was significantly improved by this method compared to MRD detection by MFC [61]. Although numerous error-correction tools are accessible, the comprehensive and precise removal of errors from sequencing data remains a challenging task. Molecular-based approaches for error correction often come with elevated computational expenses, restricting their broad applicability in clinical practice. Also, error-correction methods lack systematic comparison and standardization, limiting the value of error-corrected NGS results for MRD detection [62]. 

NGS is therefore currently not recommended to be used as a single MRD detection method. Additional investigations are needed to differentiate mutations that signal the presence of persisting AML from abnormalities associated with clonal hematopoiesis. Precise targets for measuring molecular minimal residual disease (MRD) by next generation sequencing have not been definitively established. It is known that the mere detection of a mutation in genes such as *DNMT3A*, *TET2*, or *ASXL1* does not necessarily indicate the presence of residual leukemic cells, but is commonly found in age-related clonal hematopoiesis and should not be considered as MRD [5]. Furthermore, other mutations detected via NGS in genes like *GATA2* or *RUNX1* might represent the presence of germline predisposition syndromes, which further complicates molecular MRD detection by NGS [5]. Novel approaches that attempt to overcome at least some of these limitations are discussed in the next chapter.

### 2.4. Novel MRD Detection Approaches

#### 2.4.1. Alternative Combinatorial NGS Approaches to Minimize a Potential Interference of Clonal Hematopoiesis with MRD Detection

Clonal hematopoiesis (CH) is characterized by the presence of somatic mutations of hematopoietic stem cells in the BM [63]. The presence of CH does not necessarily represent malignancy, as mutations are also found in normal cells at low VAFs without showing changes in blood counts or symptoms of hematologic disease [64,65,66]. However, patients with a presence of clonal hematopoiesis are at higher risk for the later development of myeloid malignancies [67,68], especially after having received chemotherapy for non-hematologic malignancies [69,70,71]. Mutations representing CH therefore frequently occur as first molecular aberrations prior to the onset of AML and can persist after the patient is in complete remission without representing residual leukemic cells [65]. This fact complicates post-therapeutic MRD assessment by NGS, as it is often been proven difficult to identify true leukemic mutations, even if common CH-associated mutations in *DNMT3A*, *TET2*, and *ASXL1* are excluded for MRD analysis [72,73]. It is known that about half of AML patients show persistent CH after induction chemotherapy. In contrast, alloSCT leads to the complete replacement of a patient’s hematopoiesis, resulting in loss of all initially present mutations [74].

To date, while conventional flow cytometry has been unable to differentiate between leukemic cells and clonal hematopoiesis, there have been recent approaches to overcome this limitation. Robinson et al. developed a novel multiplex single-cell MRD (scMRD) assay that combines enrichment of the targeted malignant population by MFC with integrated scDNA sequencing and immunophenotyping. Their results showed remarkably similar immunophenotypes between unmutated cells and cells harboring *DNMT3A* mutations, indicating that clonal hematopoiesis may not be associated with overtly abnormal surface protein expression. In comparison, cells harboring a leukemic mutation such as *NPM1* or *IDH2* in addition to *DNMT3A* showed consistently aberrant immunophenotypes. These findings emphasize that a combined genomic and immunophenotypic MRD analysis might have the capacity to differentiate between clonal hematopoiesis/preleukemic states and leukemic clones [75].

Although not enabling full discrimination of CH from true residual leukemic cells, we and others have developed a combinatorial MRD approach of MFC-based (leukemic) cell enrichment followed by targeted NGS of genes recurrently mutated in AML. Stasik et al. combined immunomagnetic pre-enrichment and MFC-based isolation of CD34^+^ cells with error-reduced targeted NGS. Applying this approach in samples from patients after alloSCT allowed prediction of molecular relapse with high sensitivity and specificity, even from peripheral blood [76]. In addition, this combined method enabled the detection of MRD positivity significantly earlier than conventional methods. Our group used a specifically designed combinatorial antibody panel targeting surface markers CD117, CD123, CLL-1, and TIM3 for enrichment of leukemic cells. NGS analysis on the sorted cell population then allowed highly sensitive detection of MRD in remission BM samples after induction chemotherapy, predicting relapse [77]. Slade et al. described an MRD detection approach performing whole-exon sequencing (WES) and targeted error-corrected sequencing (MyeloSeq). They performed WES on tumor and normal cells from 30 patients with AML. After the patients achieved complete remission, MRD was assessed by repeat WES and MyeloSeq. MRD positivity following induction therapy was found in the majority of patients using either MyeloSeq or WES. In some patients, comprehensive WES analysis allowed the differentiation of persistent leukemia from residual clonal hematopoiesis by identifying passenger mutations. However, the baseline mutational status was essential for proper interpretation [78].

While these described approaches may help to facilitate the discrimination of CH-associated from true-leukemic mutations detected by NGS in remission BM after chemotherapy, targeted NGS is already very helpful in early prediction of relapse after SCT. As mentioned above, successful treatment with alloSCT is expected to eliminate all myeloid mutations present earlier in the BM. Persistence or re-occurrence of mutations even in CH-associated genes, such as *DNMT3A*, *TET2*, or *ASXL1*, are therefore indicative of positive MRD, and therefore predictive of relapse as shown in some [74] but not all studies [15,79].

An infrequent but severe complication following alloSCT is the development of leukemia originating from transplanted donor stem cells, known as donor cell leukemia. The main risk factor includes predisposing germline mutations in donors, leading to the transmission of hematopoietic stem cells carrying these abnormalities to the recipient. To date, the transmission of germline variants in *DDX41*, *GATA2*, *CEBPA*, *RUNX1*, and *FANCD2* after alloSCT have been described. Interestingly, donor-derived germline mutations in *DNMT3A* have also been associated with the development of donor cell leukemia, however the current evidence is deemed insufficient to formulate clinical practice recommendations or rule out affected donors [80].

#### 2.4.2. Liquid Biopsy

Another new approach being investigated for MRD detection is analysis by liquid biopsy from blood sampling. Via liquid biopsy, components of a patient’s tumor can be isolated from the blood [81,82]. Liquid biopsies are useful for developing personalized therapies [83] and assessing treatment responses, but could also be useful in quantifying MRD in hematologic malignancies [84,85]. For MRD detection in AML, various methods using liquid biopsy exist, including detection of circulating tumor cells (CTCs), circulating RNAs (cRNAs), and circulating tumor DNA (ctDNA) [86]. In AML patients, MRD detection using liquid biopsy to detect CTCs from the collected blood sample has already been shown to significantly predict relapse-free survival [87]. A major advantage of this technique is that it would render invasive BM aspirations for MRD detection obsolete. ctDNA has undergone thorough research and has been validated as the most established and reliable source for liquid biopsies in solid tumors [88]. ctDNA can be described as fragmented cell-free DNA that is found in blood samples of patients with malignant diseases. While cell-free DNA (cfDNA) can also be found in healthy individuals as a result of apoptotic processes, the amount in cancer patients is significantly increased, due to the higher turnover rate in malignant cells. ctDNA can also be detected using NGS, and higher cfDNA concentrations have been linked to positive MRD status in AML and higher relapse rates [86]. However, limitations of cfDNA analysis for MRD assessment exist. Collected samples have to be processed almost immediately to avoid white blood cell lysis, making sample collection challenging [89]. Furthermore, samples could be either falsely negative by missing leukemic subclones with a low VAF of mutations or falsely positive due to detection of mutations related to clonal hematopoiesis [90].

#### 2.4.3. Digital PCR

A newer but still exploratory technique is represented by digital PCR (dPCR), which has the ability to quantify molecules of interest in absolute numbers normalized to the analyzed amount of blood or BM sample [91]. The process involves the initial partition of the template of interest into different compartments, and the ultimate analysis relies on thousands of individual measurements. By allowing absolute quantification, dPCR is easier to standardize and also can offer greater sensitivity than qPCR, especially when quantifying diseases at extremely low levels [19,92]. Limitations of dPCR include high variability in detectability of targets according to sample material (PB vs. BM) and not all different types of each mutation can be covered. For example, for *NPM1* only the most common mutations (Type A, B, and D) can be detected, possibly resulting in false negative MRD testing if a different *NPM1* mutation was found by NGS at the time of diagnosis [6]. However, given its high potential for interlaboratory standardization, dPCR might be used for MRD detection of distinct, more common AML-specific mutations in the future.

## 3. Impact of Residual Disease in Therapeutic Decisions in AML

Despite the significant prognostic impact of the MRD status in AML patients, there is still only a limited number of studies that tested the incorporation of the MRD status on therapeutic decision making [93]. A summary of these studies can be found in Table 2. In the current 2021 ELN recommendations regarding the clinical implementation of MRD diagnostics, it is advised that MRD-positive AML patients should be referred to individualized treatment approaches, preferably within the context of clinical trials. The ELN recommendations focus on patients with persistent, clinically relevant evidence of fusion genes or an *NPM1* mutation in qPCR, as well as such patients with molecular relapse [5]. As *NPM1* mutations can be monitored with high sensitivity using qPCR, various studies have been published where clinical decision-making has implemented the persistence or re-occurrence of *NPM1* mutation positivity after treatment [94,95,96,97].

In a group of 110 AML patients with *NPM1* mutations, Bataller et al. revealed that those experiencing a molecular relapse derived significant advantages from immediate therapy. Patients who received treatment as soon as molecular relapse was detected, achieved an impressive 2-year survival rate of 80%. In contrast, the overall survival of patients who were treated only upon the occurrence of a morphological relapse was notably lower with around 40% attaining 2-year survival. These findings emphasize the potential benefits of regular monitoring for MRD recurrence and the importance of early therapeutic intervention when a molecular relapse is detected [95]. Also, AML patients with an *NPM1* mutation and less than 4-log reduction in PB-MRD after induction chemotherapy showed a significantly higher relapse incidence and shorter OS if they did not receive alloSCT [98]. Tiong et al. also investigated the use of preemptive therapy in patients with a persistent or recurrent *NPM1* mutation after therapy. It was shown that preemptive therapy led to significantly prolonged relapse-free survival. Preemptive therapy consisted mainly of either FLAG-based chemotherapy or venetoclax plus low-dose cytarabine and immediate alloSCT. Of the ten patients who did not receive salvage therapy, the median time from molecular failure until morphologic relapse was just 21 days [94]. In their retrospective analysis of MRD results of the AMLSG 09-09 trial including *NPM1* mutated AML patients who received chemotherapy (standard arm) or chemotherapy + gemtuzumab ozogamycin (GO), Schwoerer et al. were able to show a significant increase in MRD-negative patients in the intervention arm receiving GO, which resulted in a statistically significant reduction of the cumulative incidence of relapse [96].

Furthermore, Short et al. recently discussed their data of a retrospective study on early intervention after MRD recurrence in AML patients. Their study indicated that AML patients who experience MRD recurrence after an initial MRD-negative remission as detected by MFC face a highly increased risk of morphological relapse. They found that implementing therapeutic measures such as altering the treatment regimen and/or considering alloSCT in MRD-positive patients led to improved outcomes and even durable remissions for some patients. This strongly suggests that MRD recurrence in AML indicates a state of imminent relapse for the majority of patients and MRD-guided changes in therapy can markedly improve survival [99]. In line with these findings, Puckrin et al. found that molecular monitoring for disease re-occurrence every three months in CBF-AML left an insufficient timeframe for intervention to prevent morphological relapse [100]. Therefore, the optimal intervals for MRD testing as well as the eventual impact of preemptive early changes in the therapeutic approach still need to be confirmed in prospective clinical trials.

Another impact of MRD testing on therapy has emerged from data showing that intermediate risk AML patients (according to ELN criteria) displaying a negative MRD status after two chemotherapy cycles can be considered for conventional consolidation therapy instead of alloSCT [5] without affecting overall survival. For example, Venditti et al. allocated 61 intermediate-risk patients to autologous or allogeneic SCT based on their MRD status with MRD-negative patients undergoing autologous SCT. They showed that there was no difference in 2-year OS and PFS between the MRD-positive and MRD-negative cohorts, indicating that the increased toxicity and morbidity associated with alloSCT can be spared in this group [101]. Those results are supported by a retrospective analysis of intermediate risk AML patients of the HOVON-SAKK132 trial, who either received alloSCT or conventional consolidation (autologous SCT or chemotherapy) depending on their MRD status. Their results showed no difference in relapse rates in intermediate risk patients between the group that received alloSCT and those who did not, while patients without alloSCT experienced significantly less treatment toxicity [102]. The same outcome for intermediate risk patients was observed in a recent study by Han et al. [103]. The approach to omit alloSCT in MRD-negative, intermediate risk AML patients is currently prospectively tested in a pragmatic randomized trial led by members of the ELN–DAVID AML MRD Working Group.

The significance of the MRD status in the context of alloSCT has been investigated in multiple trials. Results consistently indicate that MRD positivity before transplantation is associated with a poorer outcome [33,101,104,105,106,107,108,109,110]. Importantly, Hourigan et al. showed that patients who tested MRD-positive immediately before alloSCT had an increased risk of relapse when subjected to reduced-intensity conditioning (RIC) compared to myeloablative conditioning (MAC). Even when the increased transplant-associated mortality with MAC conditioning was taken into account, MRD-positive patients still showed improved overall survival after receiving myeloablative chemotherapy [105]. These results were supported by findings of Gilleece et al., who performed a retrospective study including more than 2200 patients using the European Society for Blood and Marrow Transplantation (EBMT) registry. All patients were in first complete remission and received alloSCT. Patients transplanted with an MRD-positive status who received myeloablative conditioning showed better leukemia-free survival, lower relapse incidence, and a tendency toward improved overall survival for individuals under the age of 50 years when compared to reduced-intensity conditioning. However, there was no advantage to using MAC regimens for patients aged 50 or older due to increased toxicity. Conversely, patients who achieved MRD negativity before transplantation, did not show better outcomes after using MAC as compared to RIC conditioning [106]. However, based on the results of the mentioned studies, it is now general practice that for patients with MRD-positive disease status, alloSCT with MAC conditioning remains the therapy of choice, as results for patients with reduced intensity regimens or no alloSCT at all remain poor [111,112]. 

The MRD status could also play another important clinical role in guiding preemptive therapy in patients with *FLT3*-mutated AML, as there are now available several FLT3 inhibitors, such as gilteritinib and quizartinib, or other multikinase-inhibitors that also target FLT3, such as midostaurin and sorafenib [113,114]. Othman et al. included 56 AML patients with an *FLT3* mutation experiencing molecular failure in their study. Although half of the patients had previously been treated with midostaurin, treatment with a novel FLT3 inhibitor resulted in 60% of patients obtaining a molecular response and 45% even achieving MRD negativity. While showing low toxicity, this treatment approach was able to bridge 22 patients to allogeneic SCT. Furthermore, Burchert et al. were able to show an improved relapse-free survival for *FLT3* mutation-positive patients in CR receiving sorafenib as maintenance therapy after alloSCT [115]. In the recent phase III MORPHO trial by Levis et al., 356 patients with *FLT3-ITD* AML underwent alloSCT and were either assigned to a maintenance treatment with Gilteritinib or placebo over a period of two years. While there was no significant difference in relapse-free survival between the two cohorts, a significant advantage was observed in the subgroup of patients with detectable MRD before SCT receiving maintenance therapy with gilteritinib [116]. All these results are promising and need to be further evaluated in prospective studies [117].

**Table 2 biomedicines-12-00599-t002:** Results of published studies implementing or evaluating MRD status in therapeutic decisions in AML.

References	Study Design	Patient Population	MRD Detection Method	Impact of MRD Results
Bataller et al., 2020 [95]	prospective multicentric study (CETLAM-12 protocol)	- 110 AML patients with *NPM1* mutation	RT-qPCR of *NPM1* mutation from sequential BM samples MRD positivity threshold was *NPM1/ABL1* > 0.05	OS according to timepoint of salvage therapy: Treatment at time of molecular relapse: 2-year OS: 80% Treatment at time of hematologic relapse: 2-year OS: 40%
Tiong et al., 2021 [94]	prospective study	- 100 *NPM1* mutated AML patients with persistent *NPM1* mutation after >2 cycles of chemotherapy or at end of treatment	RT-qPCR of *NPM1* from BM samples	RFS among patients with molecular failure (43 patients): 10.1 months (preemptive therapy) vs. 0.7 months (no preemptive therapy)
Short et al., 2022 [99]	retrospective analysis	- 740 AML patients (CBF leukemias excluded) that achieved MRD negativity in first remission - 55 patients with MRD recurrence were included for further analysis	MFC of BM samples MRD recurrence was defined as any newly detectable MRD by MFC after at least one negative BM sample	RFS according to intervention (alloSCT or change in chemotherapy) or no intervention at time of MRD recurrence: RFS 2.8/OS 10.9 months with no intervention RFS 14.9/OS 36.1 months with intervention
Venditti et al., 2019 [101]	prospective trial	- 361 de novo AML patients achieving CR after intensive chemotherapy - 78 intermediate risk patients were assessed to alloSCT if MRD-positive (43) or autoSCT if MRD-negative (35)	MFC of BM and PB samples acquired after first consolidation MRD positivity was defined as >3.5 × 10^−4^ residual leukemic cells	Survival estimates for intermediate risk patients assigned to auto or alloSCT according to MRD status: 2-year OS 79% (MRD-negative) vs. 70% (MRD-positive) 2-year DFS 61% (MRD-negative) vs. 67% (MRD-positive) There was no significant difference in 2-year OS and DFS between MRD-positive and -negative patients.
Tettero et al., 2023 [102]	retrospective analysis of the HOVON-SAKK132 trial	- intermediate risk AML patients either received alloSCT if MRD-positive or conventional consolidation (auto SCT or chemotherapy) if MRD-negative	MFC (>0.1%) and qPCR of *NPM1* (>10^−4^) in BM samples	EFS after 36 months: 47% of MRD-positive vs. 54% of MRD-negative patients 5-year OS: 54% of MRD-positive vs. 65% of MRD-negative patients
Han et al., 2022 [103]	retrospective analysis	- 235 intermediate risk AML patients	MRD assessed by MFC after 1, 2, and 3 chemotherapy cycles	MRD positivity after cycle 3 was significantly associated with worse DFS and OS (43%/44%) compared to MRD-negative patients (81%/84%) AlloSCT in MRD-positive patients after cycle 3 led to lower relapse rates after 5 years, higher DFS and OS rates than conventional therapy (22.3% vs. 71.5%, 65.9% vs. 23.0%, and 67.1% vs. 23.9%, respectively). AlloSCT did not affect outcome in the MRD-negative group.
Hourigan et al., 2019 [105]	retrospective analysis of samples from a phase III clinical trial	- 190 AML patients randomly assigned to either MAC or RIC conditioning regimen before alloSCT	Ultra-deep error-corrected NGS of frozen blood samples of patients in CR taken before alloSCT was performed retrospectively	In patients with MRD negativity, similar 3-year OS rates were shown for MAC and RIC conditioning (56% vs. 63%, respectively). For MRD-positive patients, RIC was significantly associated with increased relapse risk, decreased RFS and OS, as compared to MAC.
Gilleece et al., 2018 [106]	retrospective study	- 2292 AML patients in first CR receiving alloSCT between 2000–2015	No details on methods of MRD detection available	MRD-positive patients < 50 years old that received MAC showed significantly improved outcomes as compared to RIC conditioning: 2-year relapse incidence (36.6% vs. 50.7%) OS (59.4% vs. 43.7%) MRD-negative patients < 50 years as well as all patients over the age of 50 irrespective of their MRD status did not show a benefit from MAC conditioning.
Othman et al., 2023 [117]	retrospective analysis	- 56 AML patients with *FLT3*-ITD mutation experiencing molecular failure receiving preemptive therapy with either gilteritinib, quizartinib, or sorafenib	High sensitivity NGS for *FLT3*-ITD Testing of *NPM1* mutation was done by RT-qPCR	Treatment responses of 56 included patients: 60% achieved molecular response 45% reached MRD negativity. 2-year OS was 80% with molecular EFS being 56% High sensitivity NGS for *FLT3*-ITD was shown to be a sensitive detection method for molecular failure and was able to identify patients with potential benefit from FLT3i salvage therapy.
Othman et al., 2023 [118]	retrospective analysis of the UK NCRI AML17 and AML19 studies	- 737 *NPM1*-mutated AML patients in first CR	*NPM1* MRD status in PB measured by RT-qPCR after 2 courses of induction chemotherapy	Survival benefit of alloSCT in CR1 in MRD-positive patients (3-year OS with alloSCT 61% vs. 24% without), but no survival difference in MRD-negative patients (3 year OS 79% vs. 82%).

## 4. Role of MRD Detection in Non-Intensive Treatments

As described above, the prognostic significance of achieving MRD negativity following intensive chemotherapy or alloSCT has been thoroughly analyzed [119]. However, as lower intensity regimens with promising response rates, such as the combination of the BCL-2 inhibitor, venetoclax (VEN), and hypomethylating agents (HMA) are increasingly administered in elderly or unfit patients, the role of the MRD status in these populations has become of growing interest [119,120,121,122]. In comparison to intensive treatment, data from the assessment of MRD in patients treated with low-intensity approaches are still limited (see Table 3). This limitation likely stems from the initial expectation that most unfit patients would not achieve deep remissions with the available low-intensity treatments. With the addition of VEN to HMAs such as azacitidine (AZA) and decitabine (DEC), a significant improvement in response rates and longer overall survival could be reached, raising the question of whether the achievement of MRD negativity might also be a treatment goal in this patient population [122].

Bazinet et al. analyzed the outcome of MRD-negative patients after intensive chemotherapy compared to low-intensity treatments and showed that MRD negativity was a predictor of improved outcome irrespective of treatment intensity. In patients who responded to therapy, multivariate analysis did not reveal a significant association between treatment intensity and overall survival (OS) or cumulative incidence of relapse (CIR), but the MRD status was significantly associated with OS [123]. Even for single treatment with HMA, Boddu et al. showed a reduced risk of relapse for older patients achieving an MRD-negative CR as detected by MFC [124]. Maiti et al. included 97 patients who received DEC in combination with VEN as first-line therapy and achieved a complete remission in their analysis. The MRD was analyzed by MFC and patients who achieved an MRD-negative CR after two months showed significantly longer RFS as well as OS. Also, MRD analysis at time points of one and four months after the start of therapy was shown to predict longer survival in MRD-negative patients. A clinical implication of these results could be that patients achieving an MRD-negative CR after the first cycle could already undergo dose reduction starting from the second cycle leading to less treatment-associated toxicity [119]. 

Ong et al. found that an MFC-assessed MRD threshold of 0.1% was best predictive of RFS and OS in patients treated with VEN and AZA [125]. Othman et al. assessed MRD in 76 patients with the *NPM1* mutation during the first six cycles of first-line treatment with VEN and HMA. The rates of molecular remission with MRD negativity for *NPM1* mutation were 25%, 47%, and 50% after the second, fourth, and sixth cycles, respectively, with MRD negativity being the strongest predictor of improved OS. A significantly longer event-free survival (EFS) and OS for patients achieving MRD-negative remission after treatment with VEN and AZA was also shown by Pratz et al. [126]. These results collectively underline the clinical importance of MRD assessment in low-intensity regimens [121]. Gutman et al. investigated whether patients who achieved MRD-negative CR with VEN and AZA could discontinue treatment. However, discontinuing treatment in MRD-negative patients did not have a positive impact on the duration of response, or on OS and EFS, as compared to patients who discontinued treatment without MRD guidance, suggesting that the MRD status in patients treated with VEN and AZA may not be informative to guide treatment decisions [127].

The ELN–DAVID AML MRD Working Group has proposed recommendations for MRD testing in AML patients undergoing lower-intensity regimens for future ongoing studies, but current data are not sufficient to recommend clear-cut guidelines on MRD testing in clinical practice. The working group strongly agrees that MRD is also prognostic in patients undergoing low intensity treatment and the ELN-MRD guideline for assessing MRD using MFC-MRD and molecular MRD is also applicable to those patients. Furthermore, the establishment of an LAIP by flow cytometry before the start of low intensity treatment is recommended. The recommended time points of MRD measurements of BM in low intensity treated patients include assessments after the first, second, fourth, and seventh cycle [24].

**Table 3 biomedicines-12-00599-t003:** Role of MRD detection in non-intensive treatments.

References	Study Design	Patient Population	MRD Detection Method	MRD Results
Bazinet et al., 2022 [123]	single center retrospective study	- 635 newly diagnosed AML patients responding to intensive chemotherapy (385) or Venetoclax-based low intensity treatments (250)	MFC MRD positivity was defined as presence of a population consisting of >20 cells with an aberrant immune phenotype (DfN)	Median OS of patients: MRD-positive: 13.6 months (intensive) vs. 8.1 months (low intensity) MRD-negative: 59.2 months (intensive) vs. 18.2 months (low intensity) 2-year cumulative incidence of relapse (CIR): MRD-positive: 64.2 months (intensive) vs. 59.9 months (low intensity) MRD-negative: 41.1 months (intensive) vs. 33.5 months (low intensity)
Boddu et al., 2018 [124]	retrospective analysis	- 194 patients with newly diagnosed AML (>60 years of age) treated with a hypomethylating agent - 61 achieved a CR and MRD data from BM samples was available	MFC of BM samples MRD analysis at time of first CR and 3 months post remission	From 61 patients receiving HMA 3 months after remission 25 patients were MRD-negative 36 patients were MRD-positive. 2-year CIR: 48% for MRD-negative patients 86% for MRD-positive patients No significant difference in RFS/OS in MRD-positive and -negative subgroups.
Maiti et al., 2020 [119]	retrospective analysis	97 elderly/unfit patients with newly diagnosed AML achieving CR after decitabine + venetoclax	MFC of BM samples	54% of patients achieved MRD negativity. OS/RFS according to MRD status: OS: 25.1 months (MRD-negative) vs. 7.1 months (MRD-positive) RFS: not reached (MRD-negative) vs. 5.2 months (MRD-positive)
Othman et al., 2023 [121]	retrospective analysis	76 untreated AML patients with *NPM1* mutation who achieved CR after venetoclax + HMA or low dose cytarabine	RT-qPCR	MRD negativity for *NPM1* mutation was achieved in 25%, 47%, and 50% of patients after cycles 2, 4, and 6. OS according to MRD status after cycle 4: 84% (MRD-negative) vs. 46% (MRD-positive)
Gutmann et al., 2023 [127]	clinical trial	patients > 60 years old with newly diagnosed AML ineligible to intensive therapy received venetoclax + azacitidine	MFC/ddPCR of BM samples	Patients achieving MRD negativity discontinued azacitidine and received maintenance therapy with venetoclax, while MRD-positive patients continued combination therapy. 26/42 patients (61.9%) achieved CR, with 7 patients being MRD-negative. Discontinuing treatment in MRD-negative patients did not have a positive impact on duration of response or on OS and EFS as compared to patients who discontinued treatment without MRD guidance.

The results conclusively show that attaining complete MRD-negative remission in AML patients receiving VEN-based combinations is clearly linked to improved survival. Ongoing MRD surveillance throughout treatment in low intensity regimen patient groups can be recommended and follow-up should be tailored to individual clinical characteristics. Further randomized prospective trials are needed to implement MRD detection in non-intensive regimens into clinical decision making. By collecting prospective data, MRD monitoring may contribute to clinical decisions of therapy deintensification or discontinuation in an MRD-negative AML subgroup in the future [122].

## 5. Current Implications and Shortcomings of MRD Monitoring in AML in Daily Clinical Practice

As outlined above, the assessment of MRD has become a crucial aspect in the management of AML providing highly valuable information on a patient’s individual risk of relapse [11]. In this section, we summarize the implications as well as the shortcomings of MRD assessment in current clinical practice. 

At the initial diagnosis of AML, it is crucial to perform MFC analysis of blasts, NGS of commonly mutated genes, and PCR-based screening of fusion genes such as *RUNX1::RUNX1T1*, *CBFB::MYH11*, *PML::RARA*, *KMT2A::MLLT3*, *DEK::NUP214*, or *BCR::ABL*, if the cytogenetic analysis is of poor quality. These diagnostic steps are vital for setting the baseline for future MRD detection once morphological complete remission is attained. If leukemia-specific mutations like *NPM1* or the above mentioned fusion genes are found at diagnosis, MRD detection of these aberrations by qPCR is advised after two cycles of chemotherapy, at the EOT, as well as every three months (BM) or four to six weeks (PB) during two years of the follow-up period [5]. If PCR fails to detect any of these aberrations, subsequent MFC monitoring of MRD should be employed to identify residual cells of the initially detected leukemic population [29,30]. The time point of MFC-based MRD detection with the highest prognostic value will be after the first consolidation therapy, using a cutoff of 0.1% to distinguish MRD-positive from MRD-negative patients [28,31,32,33,34,35]. MRD monitoring using MFC after EOT is currently not recommended. NGS of mutations initially detected at diagnosis, during CR, or follow-up after EOT can be done to gain additional information on patients’ molecular remission status, while performing NGS alone cannot be advised for detection of MRD [5,49,50].

There are currently only a few clear recommendations regarding MRD-directed treatment decisions and, whenever possible, the impact of the MRD status on therapeutic decisions should be tested within a clinical trial. However, MRD recurrence in AML has been shown to be indicative of an imminent state of relapse and immediate therapy without morphologic relapse should be considered [99]. After alloSCT, CH-associated mutations such as *DNMT3A*, *TET2*, or *ASXL1* have to be considered as highly suspicious for relapse or disease persistence [15,79]. Further recommendations for the implementation of MRD in clinical practice include the use of myeloablative conditioning regimens before alloSCT for MRD-positive patients [111,112], as well as considering consolidation chemotherapy over alloSCT for MRD-negative intermediate-risk patients [102,103]. In contrast, *NPM1*-mutated patients with positive MRD after two induction cycles should receive alloSCT irrespective of their initial ELN risk to improve their outcome [118].

For non-intensive treatment strategies including venetoclax + HMA, MRD analysis in CR can be informative, as MRD remains an important prognostic factor for OS in these patients as well [121,122,123,124,125,126]. However, with current knowledge, treatment discontinuation is not advised in MRD-negative patients as further prospective data on therapy de-intensification in this patient subgroup are clearly needed [127].

## 6. Conclusions

The prognostic relevance of MRD in AML patients, as detected by MFC or molecular techniques, has been proven in numerous studies. Several advanced techniques are now available for MRD detection, however a uniform approach applicable to and comparable in all AML patients is still warranted. International collaborative efforts to compare and standardize MRD measurement methods are currently under way and their results will be crucial in fortifying the clinical utility of MRD. Of note, the FDA recently accepted the achievement of an MRD lower than 0.01% in the BM as evidence for the efficacy of new drugs administered in relapsed or refractory AML patients [128]. Eventually, the advancements of MRD detection in AML may enable physicians to tailor personalized therapy with the goal of achieving and sustaining an MRD-negative complete remission while minimizing treatment toxicity in the majority of patients. However, MRD-guided treatment changes are yet to be established in AML treatment guidelines, as results from randomized phase III trials to fully support standard clinical practice are still mostly lacking. To date, MRD-negative patients with intermediate-risk AML could receive consolidating chemotherapy instead of alloSCT. Furthermore, MRD relapse after initial molecular remission can be considered a sign of imminent relapse and should result in early treatment before full hematologic relapse. Comprehensive studies implementing MRD in treatment decisions are currently ongoing, awaiting results to identify clinical situations in which patients could clearly benefit from MRD-guided therapy. Thus, implementation of standardized MRD detection in the future is highly promising to improve treatment outcomes by individualizing therapeutic approaches in AML patients.

## Figures and Tables

**Figure 1 biomedicines-12-00599-f001:**
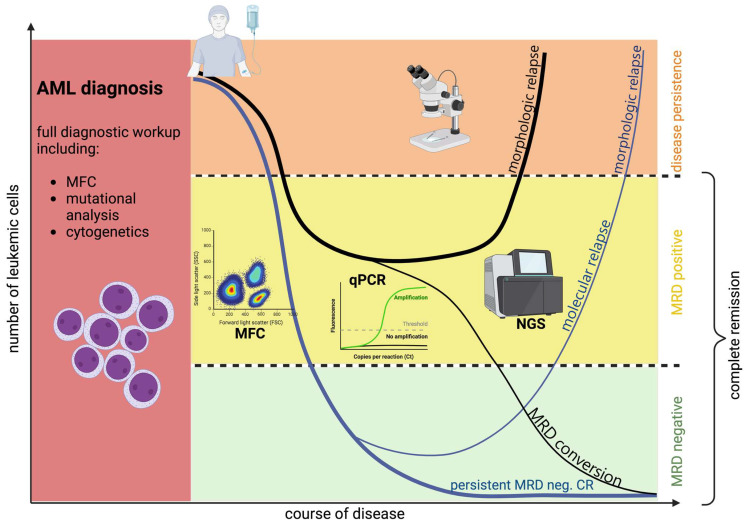
The concept of measurable residual disease in AML. Created with BioRender.com, (accessed on 2 February 2024).

**Table 1 biomedicines-12-00599-t001:** Advantages and disadvantages of current MRD detection methods for AML patients used in clinical practice.

Method	Sensitivity	Advantages	Disadvantages
qPCR	10^−5^	High sensitivity Well standardized Operator independent	Only applicable to a subset of patients (about 50%) Does not cover clonal heterogeneity/evolution Not directly quantitative
Multicolor flow cytometry (LAIP/DfN)	10^−3^–10^−4^	Applicability to nearly all patients Rapid availability of results Ease of quantification Assessment of hemodilution	Difficult to standardize Adequate interpretation requires ample experience Possible change of immune phenotype during disease course Requirement of fresh material
NGS	10^−2^–10^−4^	Broad applicability Multiple genes analyzed at once May cover clonal evolution during disease course	Not standardized yet Low sensitivity due to sequencing errors unless correction methods included Unable to distinguish true AML-specific from CH mutations in distinct cases

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
