# Peer review of "Measurable Residual Disease Detection in Acute Myeloid Leukemia: Current Challenges and Future Directions"

_biomedicines, 2024, doi:10.3390/biomedicines12030599_

Round 1
Reviewer 1 Report
Comments and Suggestions for Authors
Dear Editor-in-Chief
Thank you for inviting me to review the manuscript entitled "Measurable residual disease detection in acute myeloid leukemia: current challenges and future directions". In the present study, the authors discussed the prognostic roles of detecting MRD in AML. They demonstrated that flow cytometry, PCR, and next-generation sequencing could be utilized to recognize MRD in several studies. Although this study addressed a novel concept, some issues should be considered:
1. I recommend adding a new section regarding the shortcomings and challenges of MRD detection in the clinic.
2. The manuscript provided useful data; however, it lacks enough visual aspects. I suggest designing at least a figure to represent an overview of introductory data.
3. The mentioned studies in sections "3. Impact of residual disease in therapeutic decisions in AML" and "4. Role of MRD detection in non-intensive treatments" can be summarized in a table. Moreover, you could add new studies to the table.
4. The whole manuscript was written well, yet some mistakes are available: “due to the heterogeneity of the disease a uniform and”, “karyotype and mutational", "However only", and the like.
Comments on the Quality of English LanguageThe whole manuscript was written well, yet some mistakes are available: “due to the heterogeneity of the disease a uniform and”, “karyotype and mutational", "However only", and the like.
Author Response
We are also grateful to the reviewers for their constructive comments. We have addressed the issues raised (see below) and changed the manuscript accordingly.
- I recommend adding a new section regarding the shortcomings and challenges of MRD detection in the clinic.
Response: We have added a new section summarizing the implication and shortcomings of MRD detection in every day clinical practice (section 5, page 15)
- The manuscript provided useful data; however, it lacks enough visual aspects. I suggest designing at least a figure to represent an overview of introductory data.
Response: We have added a figure accordingly (page 2)
- The mentioned studies in sections "3. Impact of residual disease in therapeutic decisions in AML" and "4. Role of MRD detection in non-intensive treatments" can be summarized in a table. Moreover, you could add new studies to the table.
Response: We have added two tables summarizing studies mentioned in sections 3 and 4 and included new studies (table 2, page 10/11; table 3, page 14)
- The whole manuscript was written well, yet some mistakes are available: “due to the heterogeneity of the disease a uniform and”, “karyotype and mutational", "However only", and the like.
Response: We have rephrased the respective sentences accordingly.
Reviewer 2 Report
Comments and Suggestions for Authors
In this review, the authors summarized the advantages and limitations of the currently available detection methods of MRD in AML, and discussed emerging clinical implications of MRD results in the treatment of AML. This review is generally well written, and I have some minor comments on it.
1) It would be better and more readable to make a table to summarize the advantages and disadvantages of each MRD detection in terms of AML.
2) Since the outcome in elderly AML patients is much worse, what is the exact role of MRD assessment in guiding the treatment of elderly AML?
Author Response
We are also grateful to the reviewers for their constructive comments. We have addressed the issues raised (see below) and changed the manuscript accordingly.
1) It would be better and more readable to make a table to summarize the advantages and disadvantages of each MRD detection in terms of AML.
Response: We have added a table summarizing the advantages and disadvantages of each MRD detection method (table 1, page 4)
2) Since the outcome in elderly AML patients is much worse, what is the exact role of MRD assessment in guiding the treatment of elderly AML?
Response: We have clarified this topic now in a new section summarizing the implication and shortcomings of MRD detection in every day clinical practice (section 5, page 16, last paragraph)
Round 2
Reviewer 1 Report
Comments and Suggestions for Authors
Dear Editor-in-chief,
The authors satisfactorily addressed all the comments, and I assume the current manuscript is a proper one to be published.